# Analysis of Supply–Demand Relationship of Cooling Capacity of Blue–Green Landscape under the Direction of Mitigating Urban Heat Island

**Shengyu Guan [1,2], Shuang Liu [1], Xin Zhang [1], Xinlei Du [1], Zhifang Lv [1] and Haihui Hu [1,\*]**

[1]  College of Horticulture and Landscape Architecture, Northeast Agricultural University, Harbin 150030, China; guanshengyu@outlook.com (S.G.)
[2]  Architectural Design and Research Institute of Harbin Institute of Technology, Harbin 150001, China
\*  Correspondence: hljhuhaihui@neau.edu.cn

**Abstract:** Urban blue–green landscapes (UBGLs) have an important impact on the mitigation of UHIs. Clarifying the supply/demand relationship of the UBGLs' cooling effect can serve as an indicator for high-quality urban development. We established the cooling capacity supply–demand evaluation systems of UBGLs by using multi-source data and a suitable landscape mesh size. Furthermore, we utilized the coupling coordination degree (CCD) model and the linear regression equation method to explore the spatial distribution of and variation in UBGLs' cooling efficiency. The results showed the following: (1) according to the UBGL/SUHI landscape pattern index and the Pearson correlation coefficient of the land surface temperature (LST), the optimal mesh size was found to be 1200 m. (2) According to the unitary linear regression calculation, the matching of the cooling capacity supply and demand in the context of Qunli New Town showed obvious polarization; furthermore, Hanan new town and old town are more balanced than Qunli new town. (3) According to the spatiotemporal dynamic evolution of CCD, the proportion of moderate coordination- advancing cooling efficiency is the highest, reaching 35.3%. Second are moderate imbalance–hysteretic cooling efficiency (18.4%) and moderate imbalance–systematic balanced development (13.7%), with the old city highly coordinated area as the center and the coupling coordination type (gradually outward) turning into a state of serious imbalance, and then back into a state of high coordination. The findings of the investigations enriched a new viewpoint and practical scientific basis for UBGL system planning and cooling efficiency equity realizations.

**Keywords:** surface urban heat island spaces (SUHIs); urban blue–green landscapes (UBGLs); cooling efficiency; supply level; demand level; coupling coordination degree (CCD); Harbin city

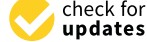



## 1. Introduction

Cities are the most densely populated regions on the planet. The drastic shift in urban material spaces and the expansion of urban temperature differences are accompanied by the rapid expansion of urban populations and urban scales. Combined with global warming, the fast-paced urbanization process leads to an increasing intensity in urban heat islands (UHIs) [1]. Artificial impervious surface and greenhouse gases enhance the surface sensible heat flux [2,3]. This has negative effects on urban regional climates [4,5], hydrological characteristics [6], physical and chemical soil characteristics [7], atmospheric environments [8], energy metabolisms [9,10], and residents' health [11–13]. The contradictions and conflicts of urban gray infrastructures and blue–green infrastructures formed by impervious surfaces have triggered a series of urban environmental problems [14], which have led to a further aggravation in the intensity of UHIs and have thus formed a vicious circle [15–17]. UHIs lead to many problems that are difficult to solve. Thus, effective measures to alleviate the effects of UHIs have become a key concern for many researchers and urban planners.

Urban blue–green landscapes (UBGLs) are ecological network systems composed of green garden spaces, urban forests, three-dimensional green spaces, urban farmlands, and bodies of water (which are an important part of urban spaces). With the advancement of UHI research, the principle of forming urban cold islands from the low reflectance of the solar radiation of UBGLs rather than the high reflectance of other urban buildings has emerged. Therefore, it is believed that this approach is capable of suppressing, counteracting, or even mitigating UHIs [18]. It should also be remarked that UBGLs have a positive impact on additional soil, water conservation, improving urban resilience, and in increasing urban well-being benefits [19]; these positive effects have become the consensus for all urban thermal environment studies [20,21]. There will be more people in areas with high temperatures, high levels of noise, as well as those with high levels of air pollution and other aerosols, especially as the urban population grows. UBGLs' inherent benefits can be used—such as reducing carbon dioxide, releasing oxygen, lowering the land surface temperature, reducing noise, reducing air aerosol content, etc.—to improve the quality of the urban ecological environment and to improve the health and happiness of city dwellers.

The cooling effect of urban blue–green spaces (UBGSs) has been studied in many fields. At present, from the perspectives of architecture, landscape architecture, and civil engineering, UBGL types or patches are usually studied by means of measurement and numerical simulation [22,23]. In addition, their local microclimates (urban park green spaces [24,25], plant communities [26,27], lake wetlands [22,28], etc.) are studied at the micro-scale level to analyze the influence of the characteristic factors of different UBGL types on LST. The research on UBGLs' cooling effect is based on GIS technology, which is combined with remote sensing image interpretation and other methods [29–31]; the cooling effect is inferred from a macro-scale analysis of landscape ecology, urban meteorology, and city planning theory. The association between the characterization of surface temperatures and the related factors of UBGLs has been studied. The research on the cooling effect has focused on the common influence of UBGLs and the construction land-cover type [32,33], geometric structures [31,34], spatial layouts [33], and surrounding environments [34] in the context of LST and against the background of urban expansion; such research found that UBGLs show different cooling effects. According to more specific research, the greater the UBGL coverage, the greater the cooling impact, and increasing UBGL areas is beneficial for reducing the effects of UHIs [35,36]. There are different studies that have suggested that the cooling impact of UBGLs becomes larger as the patch's aggregation degree increases [37]. In general, the more compact the shape of the UBGLs, the stronger the cooling effect [38,39]. Furthermore, certain research has indicated that the landscape type of the area surrounding the UBGLs is an essential component that influences the cooling effect [40,41].

Despite the fact that the breadth and depth of study on the cooling effect of UBGLs has been conducted by numerous experts, the results obtained are still not significant enough for practical urban landscape planning applications [21], and there are also problems that need to be solved. Firstly, many of the research objects for UBGLs' cooling effect concentrate on a particular type of green space or water bodies, emphasizing the need of using various cooling indicators to highlight UBGLs' cooling supply capacity. As for the cooling effect of UBGLs, we have paid much less attention to the trade-offs between the supply and demand of UBGLs. Furthermore, either directly or indirectly by UBGL alterations, human intervention processes harm the ecological environment. For UBGL arrangement alterations, there is a paucity of monitoring on cooling efficiency feedback to people. It is worth noting that many studies have revealed that the cooling ability of UBGLs is strongly affected by the landscape pattern (with noticeable scale dependence); furthermore, few studies have been conducted on the ability of different types of UBGLs to significantly mitigate UHIs under reasonable scale effects. Therefore, we also need to grasp a reasonable spatial characteristic scale in order to reflect the response of UHI to the cooling capacity of UBGLs, so as to accurately measure the spatial relationship between UBGLs and SUHIs, as well as to understand the supply and demand relationship between them.

To determine the UBGL cooling efficiency quantification process for a city, we chose Harbin, China as the research area. As a city with a great temperature difference across the four seasons and one that is accompanied by relatively large urban ecological problems, Harbin has accelerated its urban expansion in recent years. The expansion of surface urban heat island spaces (SUHIs) has resulted in increasingly major ecological and environmental challenges. In the meantime, with the constant influx of people and the rising demand for cooling, we are eager to quantify and optimize UBGL cooling efficiency imbalance areas according to the needs of supply and demand. This paper focuses on three questions: (1) how can one select the best mosaic size for the cooling effect of UBGLs? (2) how can the supply and demand relationship of UBGL's cooling capacity be measured? and (3) how can one identify and evaluate the areas with coordinated/misaligned cooling efficiencies in the process of urbanization?

From the perspective of the main urban regions with frequent human activities, in conjunction with the preceding research and the identified problems, this paper seeks to build a grid scale to quantify the changes in the cooling supply and demand benefits of UBGLs. The surface temperature of different mosaic sizes on different types of UBGLs is used to index the sensitivity of reactions and stability, as well as to determine the ideal dimension of the cooling effect. We defined the meaning of the cooling effect of the supply level (CESL) of UBGLs and the cooling effect of the demand level (CEDL) of UBGLs, in the context of SUHI landscapes, to describe the best mosaic arrangement for UBGL cooling efficiency. By revealing the spatial differences of the supply and demand level, as well as the coordination level, of UBGL cooling capacities, this paper provides a new perspective and scientific basis for UBGS layouts and UHI mitigation.

## 2. Materials and Methods

### 2.1. Overview of the Study Area

Harbin (126°8′14″~126°56′7″ E, 44°31′16″~45°55′30″ N) is the provincial capital of Heilongjiang Province. It is located in the continental monsoon climate zone, with long winters and short summers. The average winter temperature is about −20 degrees Celsius and the average summer temperature is about 23 degrees Celsius [42,43]. Harbin is not only the core area for economic development in Northeast China, but it is also an important city for population import and export, and it had a permanent population of 9.614 million [43]. Due to the large urban population base and frequent urban construction activities, a large number of green land, farmland, wetland, and other land use types have become construction land. From 2011 to 2015, urban construction was for the creation of new towns and new industrial areas. The land structure of new construction land was mainly residential land, land for roads and transportation facilities, industrial land, and land for logistics and warehousing, which all accounted for 73% of the space. In order to represent the SUHI landscapes and UBGLs directly, we chose the main urban areas (Daowai, Nangang, Xiangfang, Songbei, Pingfang, and Daoli Districts) with frequent population activities and prosperous urban construction in Harbin as the study area; these areas have a total area of 1651.44 km$^2$ (Figure 1).

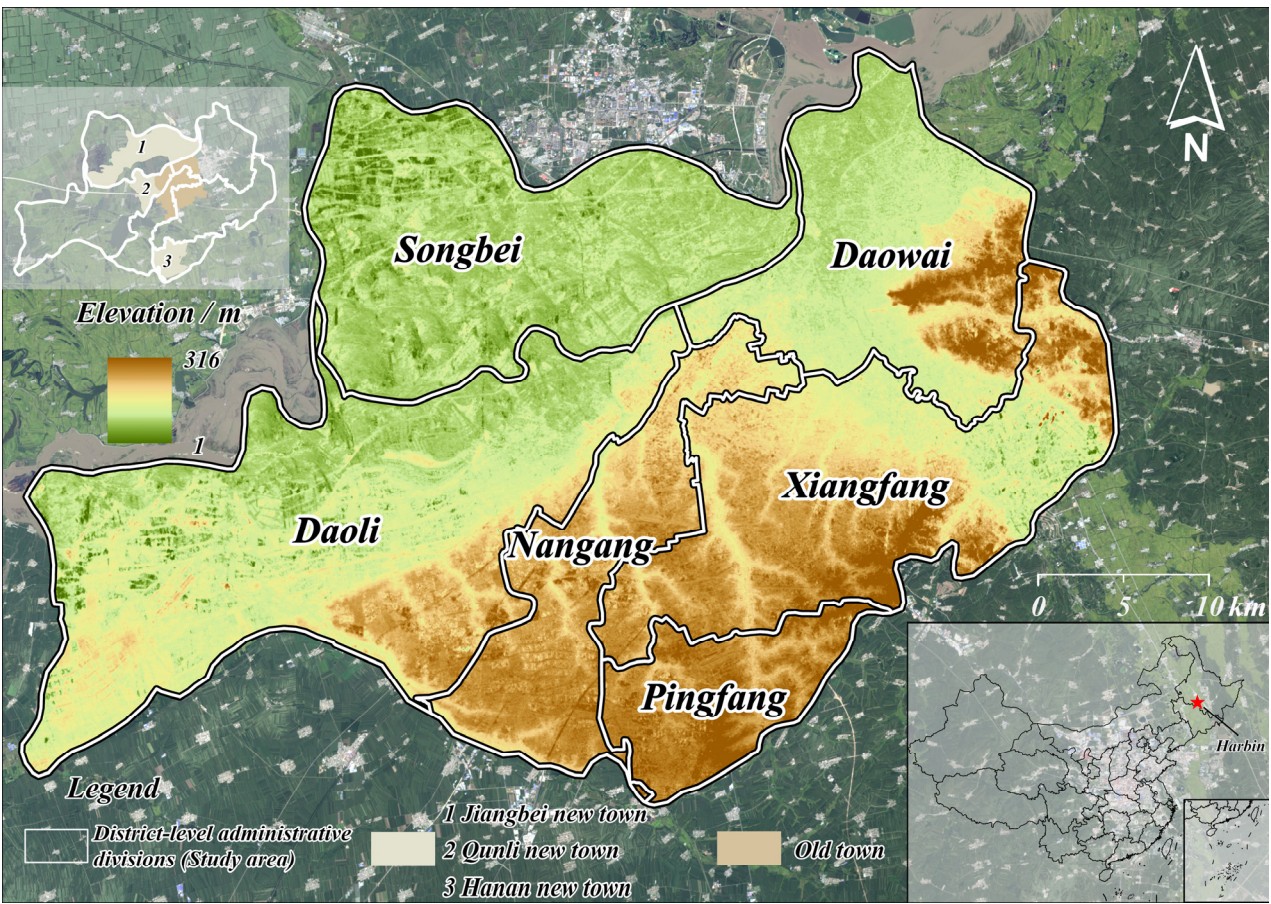

**Figure 1.** Geographical location of the main urban area of Harbin.

### 2.2. *Data Collection*

The remote sensing image data of the Landsat TM/OLI_TIRS satellite, with a spatial resolution of 30 m and cloud cover of less than 0.5%, were downloaded from the USGS website (https://earthexplorer.usgs.gov/) for the dates of 12 August 2001 (Summer), 6 September 2007 (Late Summer), 4 September 2015 (Late Summer), and 4 September 2021 (Late Summer). We used ENVI 5.3 and ArcGIS 10.6 to preprocess these images, such as radiometric calibration, atmospheric correction, and image clipping. The population density data were derived from the global Open Space Population Research data website (https://www.worldpop.org/), with a resolution of 100 m × 100 m. The night light datasets were derived from DMSP-OLS and NPP-VIIRS data (https://payneinstitute.mines.edu/eog/), the spatial resolutions of which were about 1000 m and 750 m, respectively.

### 2.3. *Methods*

#### 2.3.1. Overall Workflow

First, in this study on the main urban area of Harbin city as the research area, we obtained the spatial distribution of surface urban heat island intensity (SUHII) data and land use (LULC) data through the 2001, 2007, 2015, and 2021 remote sensing data. And we identified the grid scale with the most significant correlation between the different LULC types and LST. Then, by utilizing the mosaic units to restructure the LULC data and SUHIs, and by combining them with the cooling capacity supply–demand evaluation system, we obtained the supply–demand dataset. At last, the cooling efficiency of the UBGLs was measured from the perspectives of the coupling relationship and coordination by using the linear regression equation and the coupling coordination model (CCD), respectively (Figure 2).

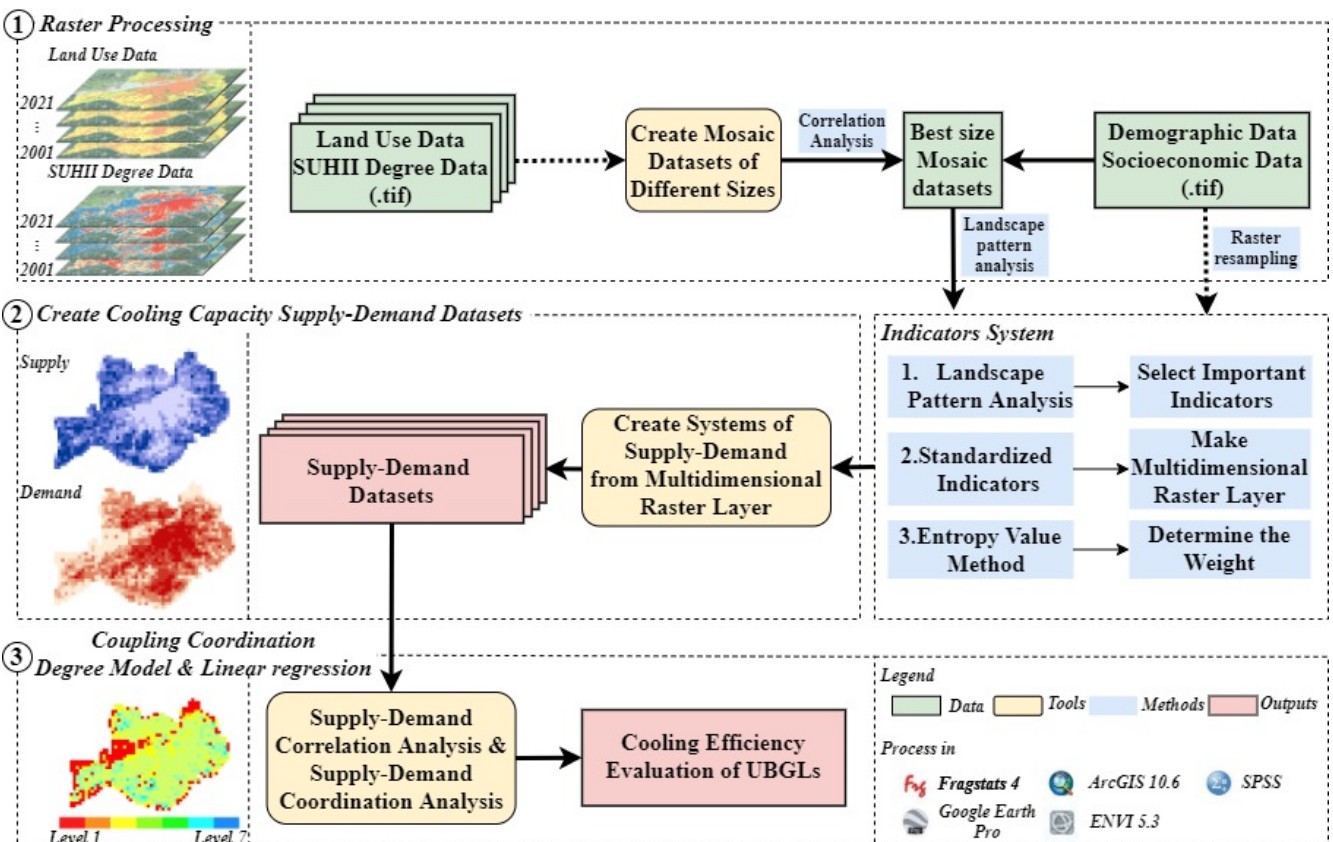

**Figure 2.** Research process.

2.3.2. Acquisition of LST and SHUIs

The Landsat remote sensing data and the radiative transfer equation (RTE) method inversion of LST were used. The retrieval precision of this method has been proven to be more accurate as its precision can reach 0.6 Celsius [28]. We corrected the Landsat remote sensing data of the four base years into a UTM coordinate system, and then obtained the LST using the RTE algorithm. The calculation process of the LST inversion is as follows:

$$L_\lambda = [\varepsilon B(Ts) + (1 - \varepsilon)\, L_{atm},\, i \downarrow]\tau + L_{atm},\, i \uparrow \qquad (1)$$

where $L_\lambda$ represents the thermal radiation intensity of the thermal infrared band; $B(Ts)$ is the ground radiance; $\varepsilon$ is the surface emissivity; $L\uparrow$ and $L\downarrow$ are the upward radiance and downward radiance, respectively; and $\tau$ represents the atmospheric transmissivity. According to Plank's law, $B(Ts)$ can be calculated as

$$B(T_S) = [L_\lambda - L_{atm}, i \uparrow -\tau(1 - \varepsilon)L_{atm}, i \downarrow]/\tau\varepsilon \qquad (2)$$

$$Ts = K_2 / ln(K_1 / B(Ts) + 1) \qquad (3)$$

where $Ts$ is the LST; the $K_1$ and $K_2$ of the Landsat 5-TM sensor data are 607.76 $\mathrm{Wm^{-2}sr^{-1}\mu m^{-1}}$ and 1260.56 K, Band 6, respectively; and the $K_1$ and $K_2$ of the Landsat 8-TIRS sensor data are 774.89 $\mathrm{Wm^{-2}sr^{-1}\mu m^{-1}}$ and 1321.08 K, Band 10, respectively. The RTE method was performed with ENVI 5.3 software.

Due to the limitation of cloud cover, there are minor differences in the seasonal effects of the remote sensing images in the four base years. Based on the results of the normalization of the LST (NLST), the mean–standard deviation (STD) method was adopted for the NLST (−1, 1) [44] to classify the thermal landscapes into five grades (Table 1). In this way, the gradient division can not only avoid the error caused by different time phases,

but can also make the temperatures more comparable. According to previous research, (5) and (4), these should be understood to be SUHIs [45].

**Table 1.** Classification standards of the land surface urban heat island intensity (SUHII) grade.

| SUHII Degree | Grading Basis |
|---|---|
| Extreme SUHI area (5) | $SUHII > \mu + STD$ |
| SUHI area (4) | $\mu + 0.5\ STD < SUHII \leq \mu + STD$ |
| Medium-temperature area (3) | $\mu - 0.5\ STD < SUHII \leq \mu + 0.5\ STD$ |
| Weak SUCI area (2) | $\mu - STD < SUHII \leq \mu - 0.5\ STD$ |
| SUCI area (1) | $SUHII \leq \mu - STD$ |

Note: $\mu$ represents the mean normalized land surface temperature (mNLST).

2.3.3. Selection of Optimal Mesh Size

In order to derive a reasonable mosaic size, we used the correlation between the different types of UBGLs and the LST (SUHII). After the radiometric calibration and atmospheric corrections of the 2001, 2007, 2015, and 2021 remote sensing images, using ENVI 5.3 software, were conducted, we used the method of combining the remote sensing parameters and the support vector machine algorithm to divide the pixels into different LULC types for the purposes of visual interpretation and supervised classification. The LULC were classified into 8 categories: cultivated land, forestland, grassland, ditch area, lake area, pond area, river area, and non-blue–green space (it includes bare land and construction area).

Since the cooling effect of UBGLs has a strong scale dependence, it is particularly important to select the optimal scale for the mosaics [34,46,47]. We needed to find an appropriate grid size. Therefore, after the mosaic grid was determined as the basic unit, sample points were generated according to the grid graph data in order to meet the needs of the statistical analysis [48]. In this study, we tried to resample the grid cell as an integral multiple of 30 m to determine the correlation between the UBGL pattern indicators and the LST. We used mosaics with side lengths of 300 m, 600 m, 900 m, 1200 m, 1500 m, and 1800 m to test the optimal mosaic scale. Through the trend and variation ranges of the Pearson correlation coefficient, the optimal grid element that was sensitive and stable to the response degree of LST to the UBGL/land use landscape type was determined; thus, the next stage of the research was ready to begin.

2.3.4. Quantitative Evaluation of CESL and CEDL

In this paper, the cooling efficiency of the UBGLs depends on the coordination between the UBGL cooling effect supply level (CESL) and the demand level of the SUHI for the UBGL cooling effect (CEDL). In order to comprehensively reflect the distribution characteristics of the landscape patterns in the main urban areas, and to effectively reduce the information redundancy so as to better quantify the spatial differentiation of the landscape spatial pattern, the experience of previous studies on UBGL/SUHI and the existence of the collinearity between certain indicators were combined. Based on the grid–cell scale, we selected six landscape pattern indicators for research: PD, LSI, AREA_MN, COHESION, ENN_MN, and AI (Table 2).

(1)　　Construction of the CESL evaluation system

The CESL reflects the cooling capacity provided by UBGLs to relieve SUHI, which depends on the UBGL conditions, as well as the LULC. Combined with the actual situation of the study area, according to the correlation analysis results between the different UBGL types and the LST (Table 3), the indicators with statistical significance were screened as the CESL indicators. Therefore, the comprehensive index system of CESL consists of 2 main indicators and 5 secondary indicators, namely AREA_MN, COHESION, AI, ENN_MN, and PLAND.

**Table 2.** The selected influencing factors in this study.

| Landscape Pattern Index | Formula | Definition |
|---|---|---|
| Patch density (PD) | $\frac{1}{A}\sum\limits_{i=1}^{M} N_i$ | The number of landscape patches per unit area |
| Landscape shape index (LSI) | $0.25E\sqrt{A}$ | The shape index of landscape patches |
| Mean patch area (AREA_MN) | $A/N$ | The average value of patch area of a certain type of landscape |
| Patch cohesion index (COHESION) | $\left[1 - \dfrac{\sum\limits_{j=1}^{n} p_{ij}{}^*}{\sum\limits_{j=1}^{n} p_{ij}{}^*\sqrt{a_{ij}{}^*}}\right]\left[1 - \dfrac{1}{\sqrt{z}}\right]^{-1} \times 100$ | Physical connectivity of the same type of plaque |
| Euclidean nearest neighbor index (ENN_MN) | $h_{ij}$ | The dispersion degree of patch distance of the same type |
| Aggregation index (AI) | $100a_{ij}/\max(a_{ij})$ | The degree of landscape patches gathered and connected |

**Table 3.** Summary of Pearson correlation between the UBGL pattern index and the LST.

| Type | Landscape Pattern Index | 2001 | 2007 | 2015 | 2021 |
|---|---|---|---|---|---|
| UGL | PD | −0.020 | −0.048 | −0.035 | 0.030 |
| | LSI | 0.187 ** | −0.001 | −0.006 | 0.432 ** |
| | AREA_MN | −0.446 ** | −0.430 ** | −0.343 ** | −0.337 ** |
| | COHESION | −0.603 ** | −0.639 ** | −0.638 ** | −0.120 ** |
| | ENN_MN | 0.353 ** | 0.412 ** | 0.403 ** | 0.048 |
| | AI | −0.708 ** | −0.589 ** | −0.635 ** | −0.316 ** |
| UBL | PD | −0.295 ** | −0.201 ** | −0.174 ** | −0.122 ** |
| | LSI | −0.198 ** | −0.142 ** | −0.153 ** | −0.234 ** |
| | AREA_MN | 0.040 | −0.025 | −0.202 ** | −0.532 ** |
| | COHESION | 0.078 | 0.125 * | −0.130 ** | −0.488 ** |
| | ENN_MN | 0.086 | −0.047 | 0.035 | 0.225 ** |
| | AI | 0.193 ** | 0.151 ** | −0.046 | −0.377 ** |
| UBGL | PD | −0.025 ** | −0.056 * | −0.048 | 0.022 |
| | LSI | −0.146 ** | −0.026 | −0.041 | 0.361 ** |
| | AREA_MN | −0.443 ** | −0.422 ** | −0.366 ** | −0.476 ** |
| | COHESION | −0.613 ** | −0.656 ** | −0.752 ** | −0.640 ** |
| | ENN_MN | 0.413 ** | 0.480 ** | 0.510 ** | 0.378 ** |
| | AI | −0.697 ** | −0.598 ** | −0.718 ** | −0.684 ** |

Note: ** indicates $p < 0.01$ and * indicates $p < 0.05$.

(2) Construction of the CEDL evaluation system

The CEDL reflects the level of cooling capacity required to mitigate SUHIs, which is determined by SUHI conditions, the LULC, the population, and the economic level. Similarly, according to the correlation analysis results of SUHI landscapes and the LST (Table 4), the significantly related indicators were selected as the CEDL indicators. Population density refers to the number of people per unit area, which reflects the human demand for alleviating the cooling effect of SUHIs. The intensity of economic activities and human activities reflects the financial strength of a city. The higher the regional financial strength, the stronger the measures taken to alleviate SUHI (such as the construction and

management capacity of urban green spaces and water bodies, the purchasing power of indoor cooling facilities, etc.), and the easier it is to avoid the harm caused by SUHIs [21]. Furthermore, as important indicators reflecting human activities, the urbanization level, economic activities, and night light data were normalized due to the different times of DMSP-OLS and SNPP-VIIRS data. Therefore, we selected 3 main indicators and 8 secondary indicators for SUHI landscapes, which are LSI, AREA_MN, COHESION, ENN_MN, AI, PLAND, POP_density, and DN.

**Table 4.** Pearson correlation between the SUHI landscape pattern index and the LST.

| Type | Landscape Pattern Index | 2001 | 2007 | 2015 | 2021 |
|---|---|---|---|---|---|
| SUHI | PD | −0.038 | 0.004 | −0.001 | 0.013 |
| | LSI | −0.218 ** | −0.155 ** | −0.315 ** | −0.164 ** |
| | AREA_MN | 0.525 ** | 0.827 ** | 0.872 ** | 0.853 ** |
| | COHESION | 0.242 ** | 0.525 ** | 0.486 ** | 0.508 ** |
| | ENN_MN | −0.106 ** | −0.385 ** | −0.335 ** | −0.432 ** |
| | AI | 0.238 ** | 0.506 ** | 0.498 ** | 0.479 ** |

Note: ** indicates $p < 0.01$.

(3)    Indicator processing and weight calculation

According to the above results, we calculated the average values for each index of the CESL and CEDL in each grid. Due to the different index dimensions selected, the minimum–maximum normalization method was adopted to process each index value in order to eliminate the differences in the dimensions and the orders of magnitude of each positive and negative index.

$$N+ = \frac{X - minX}{maxX - minX} \tag{4}$$

$$N- = \frac{maxX - X}{maxX - minX} \tag{5}$$

where $N_+$ is the normalized value of the positive indicator, $N_-$ is the normalized value of the negative indicator, $X$ is the original value of each indicator, $maxX$ is the maximum value of each indicator, and $minX$ is the minimum value of each indicator.

This study standardized the various indexes of comprehensive evaluation. The entropy method is an objective weighting method that determines the weight of an index according to the amount of information contained in the index value. Based on the principle of degree dispersion, the smaller the entropy of an index is, the greater the variation degree of the index value; in addition, the more information it provides, and the greater the role it plays in the comprehensive evaluation, the greater the weight of the index should be [49]. The calculation formula for this is as follows:

$$P = \frac{x}{\sum_{i=1}^{m} x} \tag{6}$$

$$e = -k\sum_{i=1}^{m} lnP, k = \frac{1}{lnm} \tag{7}$$

$$g = 1 - e \tag{8}$$

$$W = \frac{g}{\sum_{j=1}^{n} g} \tag{9}$$

where $p$ is the proportion of the $j$th index of the $i$th grid cell; $m$ is the number of indicators; $e$ is the entropy value of the $j$th index; and $g$ is the coefficient of variation of the $j$th index,

whereby the larger *g* is, the more obvious the effect of the index on the research object is, and thus the better the index. The weight of each factor in the index layer of the CESL and CEDL was calculated with the entropy method. The evaluation index system and weight calculation result *W* are as follows (Table 5).

**Table 5.** Evaluation index system of the CESL and CEDL in the study area.

| Destination Layer | Main Index | Secondary Index | Indictor Meaning | Property | Weight |
|---|---|---|---|---|---|
| CESL | UBGL landscape supply capacity level | AREA_MN (UBGLs) | Reflects the degree of UBGL patch fragmentation per unit area | Positive | 0.675 |
| | | COHESION (UBGLs) | Reflects the physical connectivity of UBGL patches | Positive | 0.019 |
| | | AI (UBGLs) | Reflects the aggregation degree of UBGL patches | Positive | 0.022 |
| | | ENN_MN (UBGLs) | Reflects the degree of dispersion between UBGL patches | Negative | 0.097 |
| | LULC supply capacity level | PLAND (UBGLs) | Reflects the size of the UBGL patch | Positive | 0.187 |
| CEDL | SUHI landscape demand capacity level | LSI (SUHIs) | Reflects the degree of complexity of the SUHI patch | Negative | 0.018 |
| | | AREA_MN (SUHIs) | Reflects the degree of SUHI patch fragmentation per unit area | Positive | 0.271 |
| | | COHESION (SUHIs) | Reflects the physical connectivity of SUHI patches | Positive | 0.009 |
| | | AI (SUHIs) | Reflects the aggregation degree of SUHI patches | Positive | 0.008 |
| | | ENN_MN (SUHIs) | Reflects the degree of dispersion between SUHI patches | Negative | 0.151 |
| | LULC demand capacity level | PLAND (SUHIs) | Reflects the size of the SUHI patch | Positive | 0.143 |
| | Population economic level | POP_density | Reflects the human demand for cooling capacities | Positive | 0.318 |
| | | DN | Reflects the economic capacity to deal with SUHIs | Negative | 0.082 |

2.3.5. The Analysis of the Relationship between the CESL and CEDL

In this study, we used a unitary linear regression analysis to study the correlation between the supply and demand for the cooling capacity of UBGLs. A regression analysis was used to assess the quantitative relationship of interdependence between the CESL and CEDL.

2.3.6. Cooling Efficiency Analysis of UBGLs

In this study, the cooling efficiency of UBGLs was calculated by using the coupling coordination degree model (CCD) approach. CCD synthesizes the coupling between two elements or systems and is an effective evaluation tool to study the degree of equilibrium development. The coupling coordination degree model was used to analyze the coupling coordination degree of the CESL and CEDL (Equations (10) and (11)) in order to judge the cooling benefit of UBGLs. The calculation formula for this is as follows:

$$CD_i = 2 \times \sqrt{(CESL_i \times CEDL_i)/(CESL_i + CEDL_i)} \tag{10}$$

$$CCD_i = \sqrt{CD_i \times (a \times CESL_i + b \times CEDL_i)} \tag{11}$$

where $CD_i$ is the coupling degree of the $CESL_i$ and $CEDL_i$ of the grid cell $i$; $CESL_i$ is the comprehensive supply evaluation index, $CEDL_i$ is the comprehensive demand evaluation index, and the distribution interval of both is [0, 1]. Further, $a$ and $b$ are the weights of the CESL and CEDL, respectively. In this study, supply and demand are regarded as equally important, so $a = b = 0.5$. The larger the value, the higher the degree of coordination.

## 3. Results

### 3.1. Best Mosaic Size

Through using the grid size of the remote sensing data and the fishing net tool of ArcGIS 10.6 software, we further set six mesh cell sizes as the above 300 m, 600 m, 900 m, 1200 m, 1500 m, and 1800 m, respectively. According to the different grid sizes, the Pearson correlation coefficient values ($p < 0.05$) and variation ranges of the average grid LST and the eight types of landscape PLAND in different years and different grid sizes were analyzed with IBM SPSS Statistics 26 (Figure 3).

Through the calculated results, the degree of correlation approaching zero was found at 600 m for all types of landscapes in 2001, and the correlation degree of each landscape was the worst at 900 m in 2007, 2015, and 2021. Moreover, the correlation coefficients of the four time nodes varied greatly in the range of 300–900 m. The correlation coefficients of the pond area landscapes and river area landscapes fluctuated significantly under different mosaic sizes, and the correlation coefficients of the two landscapes were higher at 1200 m and 1800 m, with a relatively stable trend. According to the above analysis, 1200 m is the most representative and optimal grid size for the landscape pattern analysis in this study; this was determined based on the variation range and sensitivity degree of various landscape correlation coefficients in four different base years and six different mosaic sizes.

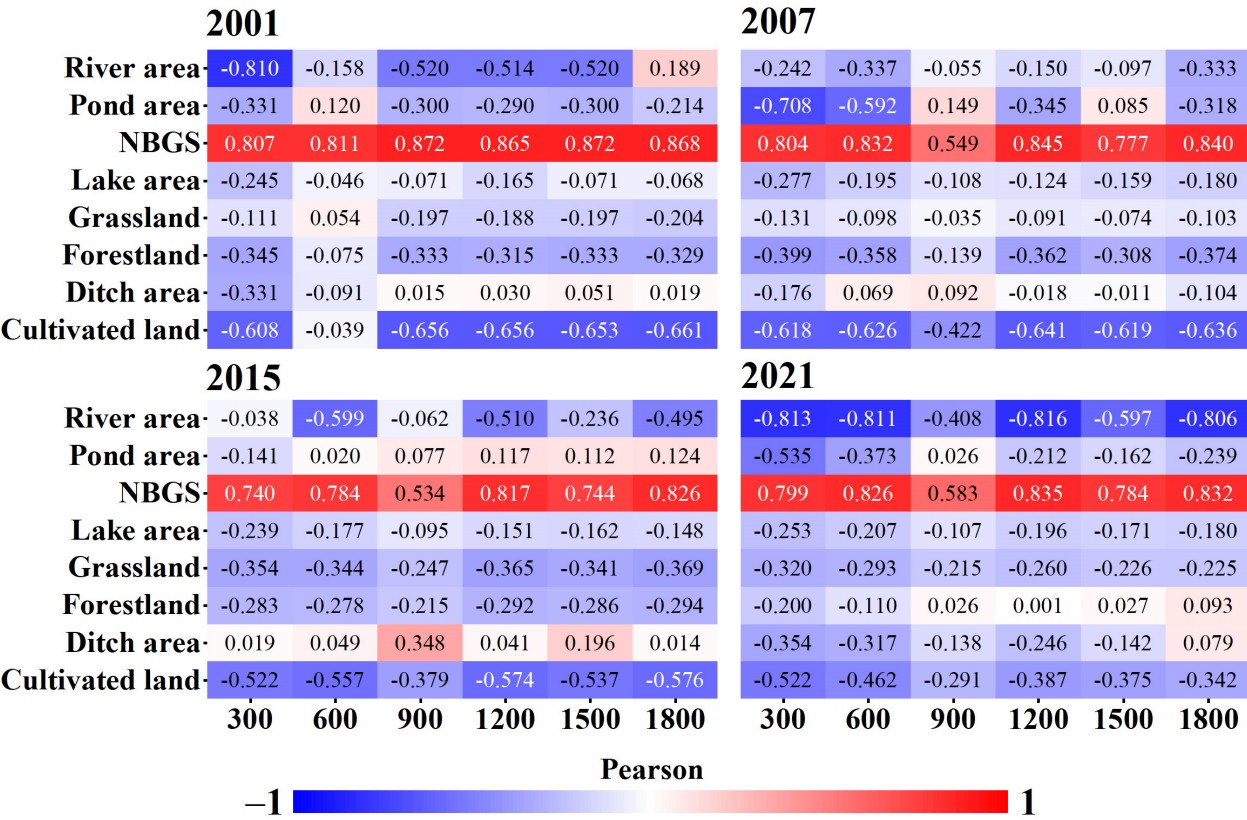

**Figure 3.** The Pearson correlation coefficient between percentage of land use and mean land surface temperatures (mLSTs) in different mosaic units.

### 3.2. Spatial Distribution and Variation of SUHIs and UBGLs

In terms of the temperature partition during 2001–2021 (Table 6), the strong SUHIs stabilized at about 21%. The proportion of normal SUHIs fluctuated little and increased steadily, with an overall increase of 1.0%. The middle temperature fluctuated greatly in the region, and showed a decreasing trend as a whole, with an overall decrease of 3.0%. The proportion of the SUCI landscapes fluctuated greatly, and the increase was 2.0%.

**Table 6.** The area ratio of each temperature zone in the study area.

| Primary Zoning | SUHII | 2001 | 2007 | 2015 | 2021 |
|---|---|---|---|---|---|
| Hot zone | 5 | 21.3% | 20.8% | 21.2% | 21.2% |
|  | 4 | 14.0% | 15.3% | 14.6% | 15.1% |
| Normal zone | 3 | 29.8% | 27.5% | 30.0% | 26.8% |
| Cold zone | 2 | 20.0% | 15.7% | 10.4% | 15.8% |
|  | 1 | 14.8% | 20.7% | 23.7% | 21.0% |

From the perspective of spatial distribution, the SUHI patches in the old town form a gradual spreading trend from the inside to the outside; at the same time, in the new town, the isolated SUHI patches, led by Songbei District, were gradually expanding, and the connectivity of SUHI patches in Pingfang District was gradually increasing and spreading toward the old town. The above situation confirms that the construction of new urban areas will lead to the formation of a SUHI region. In addition, the SUHI patches in the outskirts of Daoli District were gradually decreasing, which may be related to the policy of cultivated land protection and the trend toward afforestation in recent years (Figure 4a).

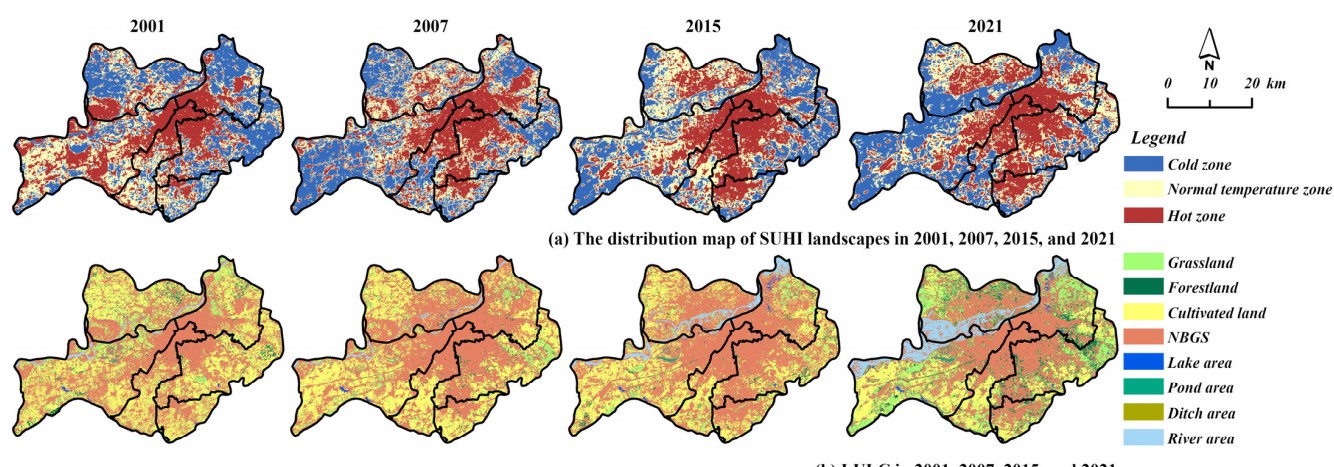

**Figure 4.** The distribution map of SUHI landscapes (**a**)/LULC types (**b**) for the main urban areas of Harbin from 2001 to 2021.

According to the spatial distribution of the LULC from 2001 to 2021 (Table 7 and Figure 4b), the cultivated land and NBGS areas were the most dominant landscapes in the urban areas, and were also the main landscape types used in land use transfer. During the studied 20 years, the cultivated land decreased by 237.28 km$^2$ and NBGS decreased by 68.41 km$^2$.

**Table 7.** Area of land use landscape types in the study area.

| Land Use Types | 2001 | | 2007 | | 2015 | | 2021 | |
|---|---|---|---|---|---|---|---|---|
| | Area (ha) | Ratio (%) | Area (ha) | Ratio (%) | Area (ha) | Ratio (%) | Area (ha) | Ratio (%) |
| Grassland | 18,172.71 | 11.00 | 8898.26 | 5.39 | 8457.88 | 5.12 | 26,348.09 | 15.95 |
| Pond area | 621.81 | 0.38 | 174.87 | 0.11 | 541.44 | 0.33 | 528.75 | 0.32 |
| NBGS | 82,768.60 | 50.12 | 92,027.47 | 55.73 | 94,994.48 | 57.52 | 75,928.03 | 45.98 |
| Cultivated land | 56,311.38 | 34.10 | 56,781.32 | 34.38 | 48,803.57 | 29.55 | 32,583.02 | 19.73 |
| Ditch area | 33.87 | 0.02 | 13.21 | 0.01 | 117.07 | 0.07 | 162.33 | 0.10 |
| River area | 2753.46 | 1.67 | 2746.98 | 1.66 | 7148.34 | 4.33 | 13,957.83 | 8.45 |
| Pond area | 353.79 | 0.21 | 400.68 | 0.24 | 722.61 | 0.44 | 888.48 | 0.54 |
| Forestland | 4128.57 | 2.50 | 4098.39 | 2.48 | 4358.80 | 2.64 | 14,747.66 | 8.93 |
| Total | 165,144.18 | 100.00 | 165,144.18 | 100.00 | 165,144.18 | 100.00 | 165,144.18 | 100.00 |

With the passage of time, the proportion of river areas increased rapidly, from 1.67% to 8.45%, and reached its highest in 2021. The forestland and grassland areas showed modest increases, 6.33% and 4.85%, respectively. Other UBGL landscape types were not found to be advantageous landscape types, and they also changed sizes for unclear reasons. It is worth noting that, combined with the spatial distribution of SUHIs and UBGLs, the scale growth led by river areas had greatly helped to form the local SUCI landscape. In 2021, the expansion of forestland and grassland in the suburban area reduced the degree of urban enclave expansion, and the expansion became scattered.

### 3.3. Spatial Distribution and Variation in the CESL, CEDL, and CCD

The results of the CESL and CEDL were divided into seven types via a natural break-point method: extremely low supply/demand (1), low supply/demand (2), relatively low supply/demand (3), medium supply/demand (4), relatively low supply/demand (5), high supply/demand (6), and extremely high supply/demand (7). And we divided the coordination types into the following seven levels: extreme dissonance (0 < CCD < 0.2), moderate dissonance (0.2 ≤ CCD < 0.3), mild dissonance (0.3 ≤ CCD < 0.4), borderline

dissonance (0.4 ≤ CCD < 0.5), based coordination (0.5 ≤ CCD < 0.6), medium coordination (0.6 ≤ CCD < 0.7), and good coordination (0.7 ≤ CCD < 1.0).

The CESL results are shown in Figure 5a–d. The high-value areas of the CESL were mainly distributed in the outer ring area (i.e., to the west of Songbei District, the west of Pingfang District, the west of Nangang District, the east of Daowai District, etc.). The population density in these areas was low, and the coverage rate of the UBGL landscape, which was mainly cultivated land, was high. The low value areas of the CESL were found in the old town center. With continuous urban expansions in the past 20 years, the supply grade of the old town and the new urban area was gradually decreasing. Meanwhile, with the expansion of the scale of SUHIs and due to the increase in the population density, the low-supply-grade areas were also gradually spreading, and the high/medium-supply areas were gradually decreasing into low-supply areas.

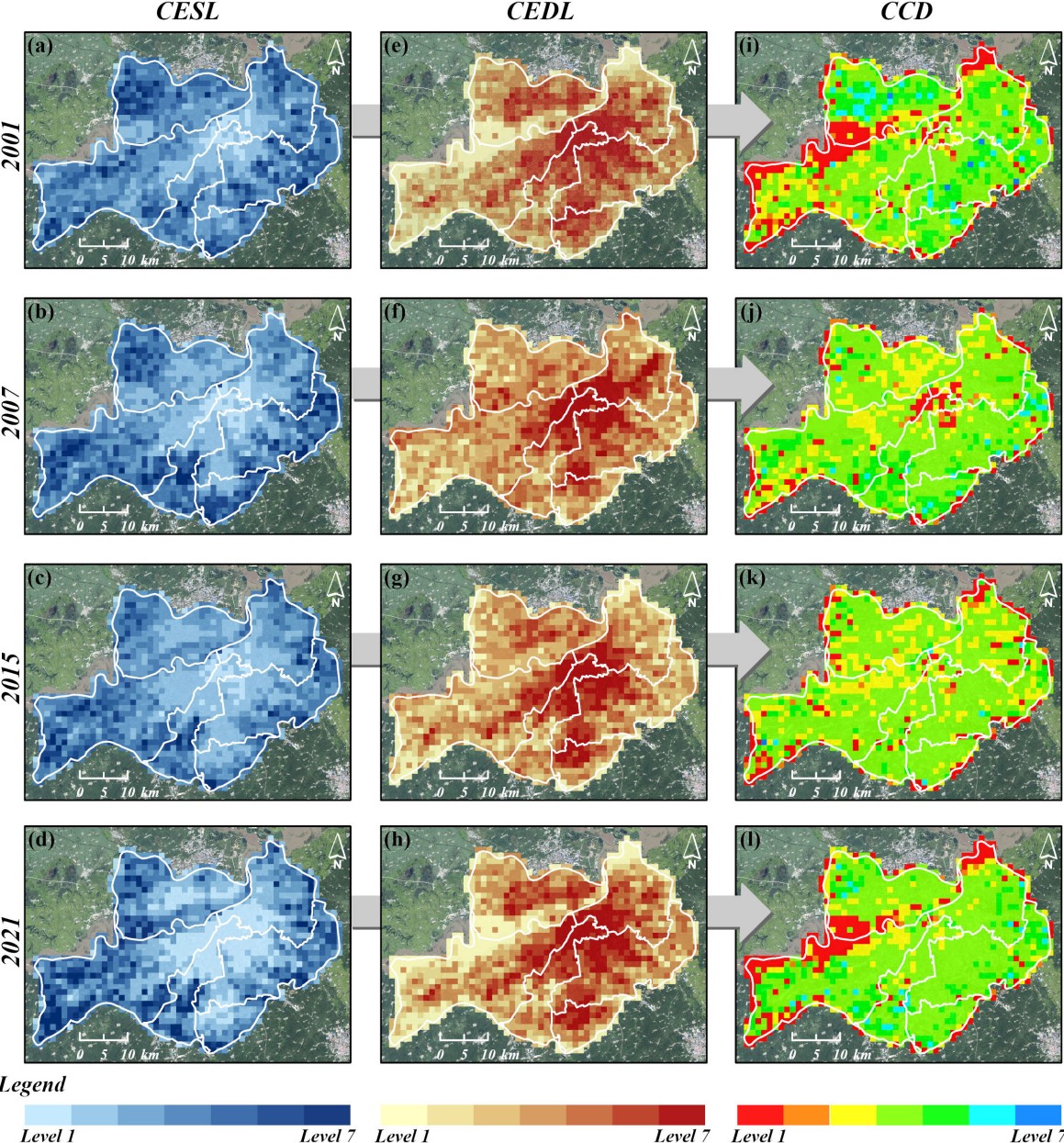

**Figure 5.** The spatial distribution of the different grades of the CESL (**a–d**), CEDL (**e–h**), and CCD (**i–l**) in 2001, 2007, 2015, and 2021.

The CEDL results are shown in Figure 5e–h. The high-value areas were mainly distributed in the construction land of the old town and the new urban area with no significant change being shown throughout the 20 years except that the CEDL was generally high in 2007. By taking the center of Songbei District, the dense area of the old town, and the center of Pingfang District as three core areas, the demand capacity gradually decreased from the inside to outside. This finally formed a situation of gradual transition between the three cores. The low CEDL value areas were mainly distributed in Songhua River Basin and to the southwest of the study area. The reason for the low demand was that the low CEDL value areas existed in rivers, wetlands, lakes, arable land, and other areas where the population density was low, and where the infrastructure construction was relatively backward. In general, in the past 20 years, the degree of demand for SUHI landscapes for UBGLs remained essentially unchanged, and the cooling capacity of UBGL landscapes in the old town and the new urban area was found to always be at a relatively high/high level.

The cooling efficiency results for CCD-based UBGLs from 2001 to 2021 are shown in Figures 5i–l and 6. The low-value areas of the CCD were mainly located in Songhua River Basin, which is southwest of the study area, and in the old town of Daoli District and Daowai District, which are scattered in the boundary areas of the study area. The high-value areas of the CCD were mainly distributed in the southern suburbs of the study area. During the 20-year study period, the CCD values first decreased and then increased in most of the areas of Songbei District. Likewise, in the upper reaches of Songhua River, the CCD value first increased and then decreased. These areas were mostly new urban construction areas and wetland construction areas, and maintained a high cooling capacity supply during the 20-year study period. The CCD value in the downtown area has steadily increased over the past 20 years, but it is still in a state of moderate dissonance or mild dissonance. In addition to the above regions, the overall regions were mainly classified as mild dissonance, borderline dissonance, and based coordination. In 2021, the based-coordination-type regions were scattered in the suburbs, while the mild dissonance and borderline dissonance regions were concentrated in the riverside zone of the Songhua River region and suburbs.

### 3.3.1. Coupling Relationship between the CESL and CEDL

Due to the large differences in the natural background and socioeconomic characteristics of the different mosaics, we specifically chose new and old urban regions with strong economic development as the typical supply and demand correlation areas in order to better understand the relationship between the CESL and CEDL in different locations. The new districts include Jiangbei New Town in Songbei District, Qunli New Town in Daoli District, and Hanan New Town in Pingfang District. The old town is located at the junction of Daoli District, Daowai District, Nangang District, and Xiangfang District.

As shown in Figure 7, only the CESL and CEDL in Jiangbei New Town had a weak positive association in 2001, whereas the CESL and CEDL in other years and regions had a negative correlation. The supply of UBGL cooling capacity in new urban areas may not meet demand due to the net increase in impervious surfaces in high-demand areas with a rapid economic development, or due to the low supply efficiency of UBGL cooling capacities in newly built urban parks and wetland parks.

Meanwhile, through Figure 7, we discovered regional disparities in the CESL and CEDL of the four typical regions. By comparing the slope of the fitted line in the four regions, we discovered that the Jiangbei region had the highest slope value, while the Qunli region had the lowest. For the past 20 years, the slope in the Hanan region was constant, ranging between 0.65 and 0.74. The phenomenon was mostly caused by the regional disparity in the background difference between the supply and demand for UBGLs. As the physical distance between the Qunli region and the old city is relatively short, (whereas the Hanan region and Jiangbei region are far), the Qunli region has more advantages in terms of urban construction, population migration, and aggregation, as well as in intensive land

use and development, which thus leads to a decline in the CESL and the high level of CEDL. When compared to the Qunli region, the UBGS resources in the Jiangbei region and Hanan region had a stronger foundation, a moderate level of development, and a more balanced supply and demand for cooling capacity. In fact, there are currently seven big urban parks and wetland parks in the Qunli region. However, in comparison to the number and size of urban parks, the urban buildings and impervious water cover (population is also part of the reason) increase the need for UHI. The aforementioned factors may contribute to the low slope of linear regression observed for Qunli New Town.

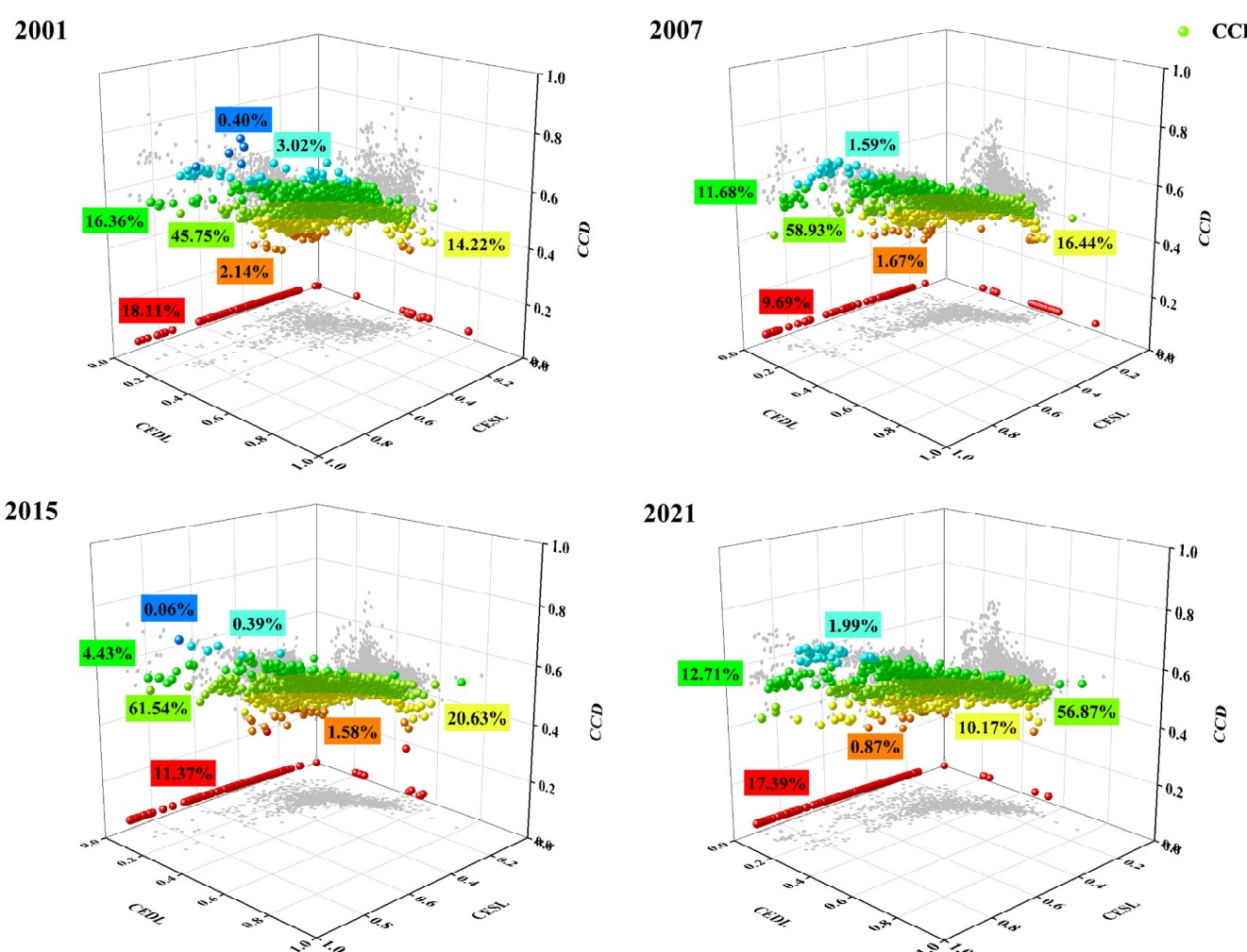

**Figure 6.** The change in the CCD spatial pattern in the main urban area of Harbin.

### 3.3.2. Spatiotemporal Dynamic Evolution of CCD

Utilizing the methods from the above traditional study, we attempted to combine the CCD values from several years of supply and demand values in order to describe the dynamic evolution and trend of the CCD in the main urban area of Harbin over the last 20 years. By using the difference between the CCD in 2001 and the CCD in 2021, as well as the difference between the CESL and CEDL, we documented the type of variation in the CCD differences from 2001 to 2021.

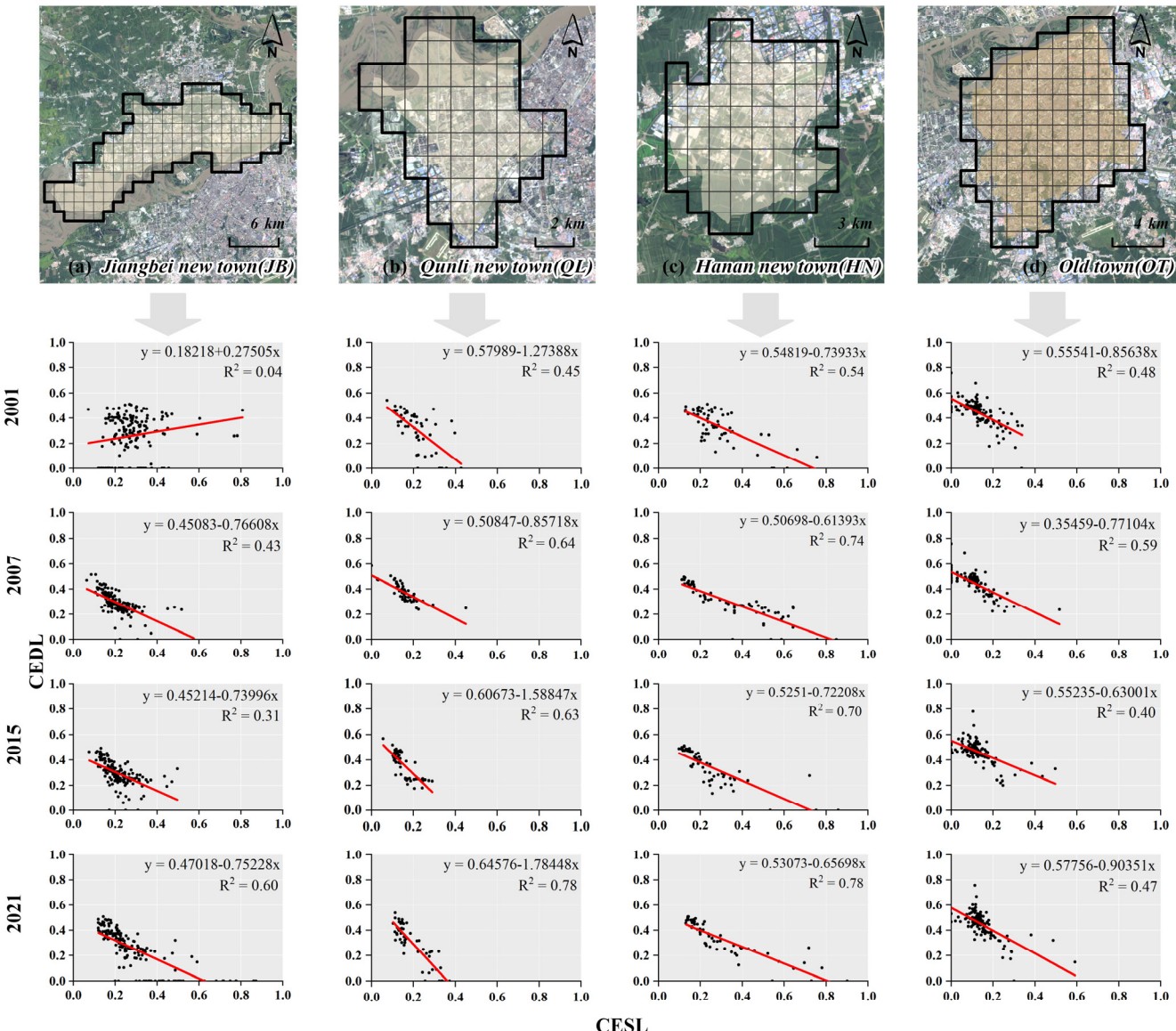

**Figure 7.** Correlation analysis of the CESL and CEDL in the three typical new towns (**a**–**c**) and the old town (**d**).

The specific types of coupling coordination are classified into three groups based on the change in the gap between the CESL and CEDL: advanced cooling capacity, balanced system development, and lagging cooling capacity. The results showed that system transformation development is the primary mode of development in the main urban area of Harbin. Figure 8 shows that the percentages of the region for advanced cooling capacity, balanced system development, and lagging cooling capacity were 48.2%, 24.1%, and 27.8%, respectively.

We further divided the data, i.e., the data of Table 8, into 12 groups based on the comparative classification of supply and demand given above, as well as by the specific types of coupled coordinated development.

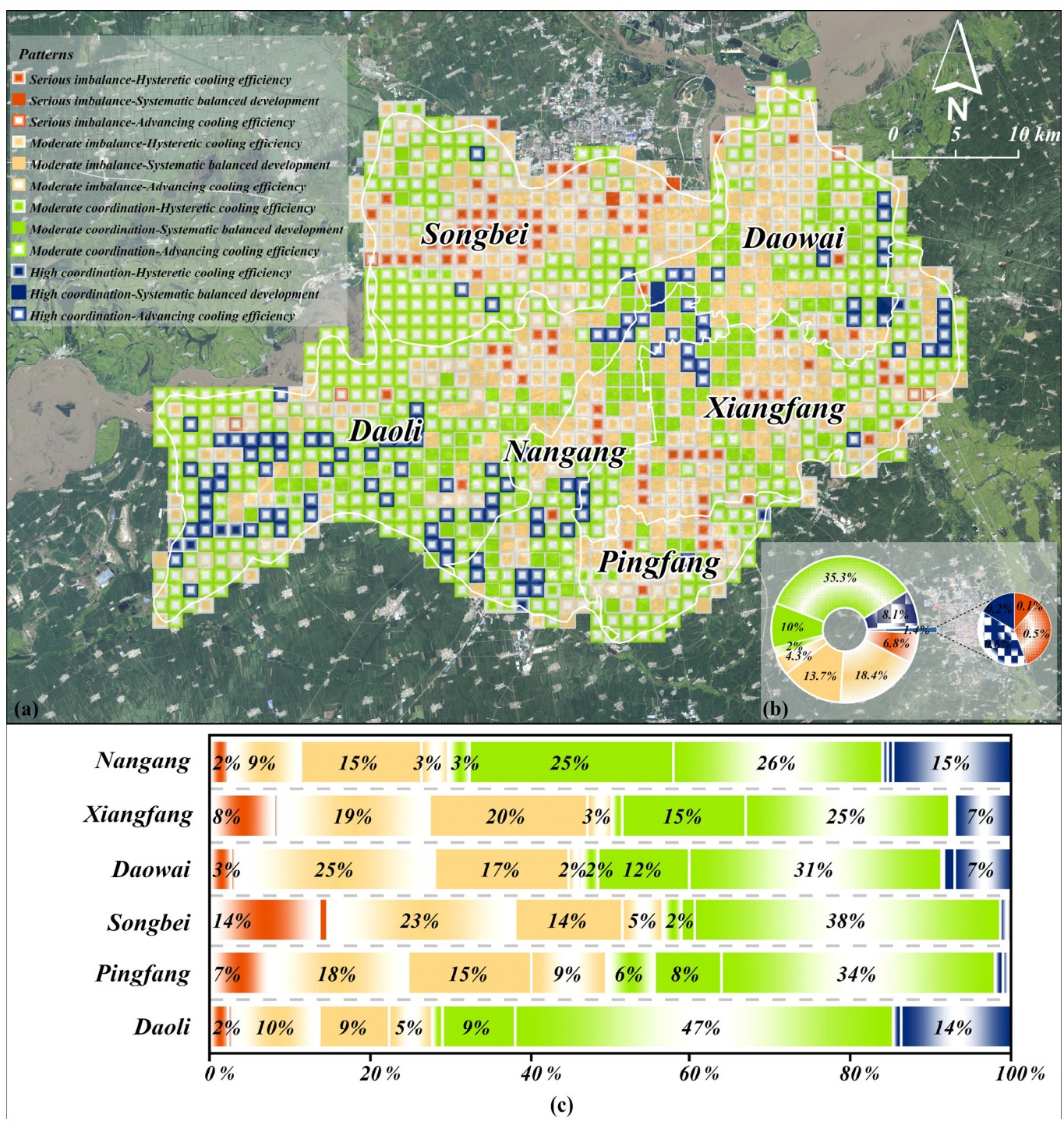

**Figure 8.** Spatial and temporal pattern of the CCD during 2001–2021 (**a**), a pie chart of the different types of the CCD (**b**), and the pattern proportion of the district-level administrative divisions (**c**).

As shown in Figure 8b, we found that the proportion of moderate-coordination-advancing cooling efficiency was the highest, reaching 35.3%. Secondly, there are three types that accounted for between 10% and 20%, namely the moderate-imbalance-hysteretic cooling efficiency (18.4%), moderate-imbalance-systematic balanced development (13.7%), and moderate-coordination-systematic balanced development (10%). The other specific types accounted for less than 10%.

**Table 8.** Classification categories for CCD dynamic evolution.

| Composite Category | Coordination Level | Subcategory | Specific Exponential Comparison | Subcategory |
|---|---|---|---|---|
| System coordinated development | $CCD_{2021} - CCD_{2001} > 0.1$ | High coordination | $\|CESL_{2021} - CEDL_{2021}\| > 0.1$ | Advancing cooling efficiency |
| | | | $\|CESL_{2021} - CEDL_{2021}\| \leq 0.1$ | Systematic balanced development |
| | | | $\|CESL_{2021} - CEDL_{2021}\| < -0.1$ | Lagging cooling efficiency |
| System transformation development | $0.1 \geq CCD_{2021} - CD_{2001} > 0$ | Moderate coordination | $\|CESL_{2021} - CEDL_{2021}\| > 0.1$ | Advancing cooling efficiency |
| | | | $\|CESL_{2021} - CEDL_{2021}\| \leq 0.1$ | Systematic balanced development |
| | | | $\|CESL_{2021} - CEDL_{2021}\| < -0.1$ | Lagging cooling efficiency |
| | $0 \geq CCD_{2021} - CCD_{2001} > -0.1$ | Moderate imbalance | $\|CESL_{2021} - CEDL_{2021}\| > 0.1$ | Advancing cooling efficiency |
| | | | $\|CESL_{2021} - CEDL_{2021}\| \leq 0.1$ | Systematic balanced development |
| | | | $\|CESL_{2021} - CEDL_{2021}\| < -0.1$ | Lagging cooling efficiency |
| System uncoordinated development | $-0.1 \geq CCD_{2021} - CCD_{2001}$ | Serious imbalance | $\|CESL_{2021} - CEDL_{2021}\| > 0.1$ | Advancing cooling efficiency |
| | | | $\|CESL_{2021} - CEDL_{2021}\| \leq 0.1$ | Systematic balanced development |
| | | | $\|CESL_{2021} - CEDL_{2021}\| < -0.1$ | Lagging cooling efficiency |

Figure 8a depicts cooling efficiency lag as the primary form of spatiotemporal dynamic coupling in Songbei District (Jiangbei New Town), which is to the north of Songhua River. The old town showed different levels of a balanced development of supply and demand. From knowing the coordination type of the old town, most areas were developed in the systematic balanced development approach. However, what merits special notice is that, with the old city's highly coordinated area as the core, the coupling coordination type gradually turned outward into a state of serious imbalance, and then back into a state of high coordination.

Figure 8c shows the proportion of the various coordination patterns in the six district-level administrative regions. In the 20 years of urban development, Pingfang District and Songbei District had the highest proportion of hysteretic cooling efficiency, i.e., 31% and 39%, respectively. The proportion of the advancing cooling efficiency area in Daoli District was the highest, reaching 66%. Nangang District had the highest proportion of systematic balanced areas, reaching 40%. These results further confirm that the large-scale construction of the new town caused a hysteresis effect in the cooling efficiency of the UBGLs, and that the cooling effect of the urban parks (wetlands) that were built in support was not significant.

## 4. Discussion

### 4.1. Quantitative Evaluation of the CESL and CEDL

UBGLs can effectively alleviate UHIs, and the spatial distribution of UBGLs also influences a UHI phenomenon to some extent. The surface heat flux increased in the study area as a result of the change in the natural cover generated by the development of the new district. Furthermore, the supply of freshly developed UBGLs does not satisfy the human need for cooling capacities, thus resulting in a breakdown in the supply and demand balance of the original UBGL cooling capacities. Therefore, it was necessary to combine the CESL and CEDL systems to reveal the spatial heterogeneity of UBGL cooling efficiency in a city, which we tested in the main urban area in Harbin. In this study, the CESL represents the supply level of the cooling capacity provided by the UBGLs in order to alleviate the SUHI, while the CEDL represents the demand level of the SUHIs for the cooling capacity of UBGLs. A description of research results shows that the CESL and the CEDL presented a strong spatial heterogeneity in the different regions. The CESL values showed a low trend in the urban areas and a high one in the suburban areas. Additionally, a low-value area was in the center, and this low-value area gradually expanded outward. Conversely, the CEDL value was high in urban areas and low in suburban areas, and the value remained high. Using the three new towns and the old town as examples, a deeper examination of the link between the CESL and CEDL facilitated the study of the UBGLs' spatial allocation and provided an accounting basis for their supplementary demand. Through the accounting results of the CESL and CEDL, the supply and demand relationship of the cooling capacity can be quantitatively evaluated, which is helpful for the spatial configuration of UBGLs; in addition, it can be adjusted according to the situation of the CESL and CEDL, which is the basic prerequisite for achieving the balance between supply and demand.

### 4.2. The Improvement in the Cooling Efficiency Enlightens UBGL Planning

We measured the relationship between the CESL and CEDL via a CCD model and a linear regression equation of one variable. The CCD's dynamic coupling characteristic was used to determine the equilibrium condition of the supply and demand in a specific area, and the cooling efficiency of the UBGLs in the cities was further evaluated based on the above method. For example, in the densely populated new and old town areas, there was a poor cooling efficiency that was caused by low CESL values. In sparsely populated suburban areas, higher CESL values also led to less desirable cooling efficiencies.

As one of the most famous cities in China, Harbin suffers from a serious imbalance in cooling efficiency. In the past 20 years, the fragmented protection and construction of Songhua River, wetland parks, and other green spaces still cannot solve the overall UHI effect. There is still a long way to go to create a green and resilient city that meets the demand of humans for cooling effects. According to the concept framework of environmental justice (EJ) and high-quality development, solving the spatial relationship between supply and demand is an important aspect for urban planners to consider. How should one improve the cooling efficiency of UBGLs? Or, in order to realize the energy transmission process between the priority intervention planning area, how should one use the low-value area of a CCD with a high supply and low demand to transport the excess energy for cooling capacities to the low-value area of a CCD with a low supply and high demand? Such problems and planning ideas help to ease the UHI effect, and can provide a new way of thinking and scientific basis.

### 4.3. Limitations and Prospects

This paper was mainly based on the perspective of a grid geographic unit and landscape pattern. In this way, the measuring of the CESL/CEDL and the relationship between them in a city as a basis for further evaluating the UBGLs' cooling efficiency was conducted. However, due to the limitation of the perspective and scale, the evaluation index is still in the preliminary stage and needs to be further improved. How to improve the cooling efficiency of UBGLs, how to mitigate the nature of SUHIs or to improve the coverage of

UBGLs, as well as how to systematically and specifically configure UBGLs to address the supply and demand mismatch have not yet been discussed. Therefore, measuring the state of supply and demand for UBGLs requires a more multidimensional interpretation in future studies. At the same time, besides starting from the perspective of landscape patterns, targeted optimizations for the supply and demand flow configurations of UBGLs should be carried out from more multidimensional perspectives. This is what the authors of this paper will further consider and solve.

## 5. Conclusions

This paper is based on the perspective of landscape patterns, as well as the combination of the Harbin city UBGL landscape index and the correlations between the LST. The best grid cell was found to be 1200 m, and the CESL and the CEDL evaluation system was established to measure the research area of the CESL and CEDL spatial relations and UBGL cooling efficiencies. As a result, the main conclusions are as follows:

(1) According to the unitary linear regression calculation, the matching of the CESL and CEDL of Qunli New Town showed obvious polarization, and the regions with high supply and low demand and low supply and high demand were mostly similar, which resulted in the lowest slope line of fitting among the four case areas. The results of Jiangbei New Town, Hanan New Town, and the old town were more balanced than those of Qunli New Town;

(2) It can be seen from the spatiotemporal dynamic evolution of the CCD that the percentages in the regions of advanced cooling capacity, balanced system development, and lagging cooling capacity were 48.2%, 24.1%, and 27.8%, respectively. The proportion of moderate-coordination-advancing cooling efficient areas was the highest, reaching 35.3%. Secondly, the moderate-imbalance-hysteretic cooling efficient areas represented 18.4%, the moderate-imbalance-systematic balanced development areas were 13.7%, and the moderate-coordination-systematic balanced development areas were 10%. In terms of spatial distribution, the old town showed different levels of balanced development for supply and demand. From the coordination types of the old town, most areas were developed with a systematic balanced development approach. What merits special notice is that, with the old city highly coordinated area as the core area, the coupling coordination type gradually turned outward into a state of serious imbalance, and then back into a state of high coordination;

(3) The extremely unbalanced areas with low supply and high demand were accompanied by a high population density and socioeconomic level, which are the main reasons for low cooling efficiencies. Therefore, the construction intensity of such areas should be controlled, the coverage of UBGLs should be emphasized, and the population size should be managed. Other major reasons for low cooling efficiency are the extremely disordered areas of high supply and low demand, high coverage rates of UBGLs, and extremely low population densities. Therefore, the degree of utilization for cooling capacities in such areas should be emphasized. In addition, the ecological corridor constructions of UBGL cooling capacity flows can be considered, and these can enter the urban area through the energy transportation of cooling capacities from other areas, which can be achieved with a macro perspective of the city.

**Author Contributions:** Conceptualization, S.G., H.H. and S.L.; methodology, H.H. and S.G.; data curation and formal analysis, S.G.; writing—original draft preparation, S.G.; writing—review and editing, H.H. and S.L.; visualization, S.G., X.Z., X.D. and Z.L.; supervision, S.L., X.Z. and H.H.; funding acquisition, H.H. All authors have read and agreed to the published version of the manuscript.

**Funding:** This work was funded by the Natural Science Foundation of Heilongjiang Province, grand number LH2021E006; the Social Science Foundation of Heilongjiang Province, grand number 19YSB098.

**Data Availability Statement:** The data used in this study can be requested from the authors.

**Conflicts of Interest:** The authors declare no conflict of interest.

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
