# Peer review of "Analysis of Supply–Demand Relationship of Cooling Capacity of Blue–Green Landscape under the Direction of Mitigating Urban Heat Island"

_sustainability, doi:10.3390/su151410919_

Round 1

Reviewer 1 Report

This manuscript (Examining) Analysis of supply-demand relationship of cooling capacity of blue-green landscape under the direction of mitigating urban heat island is prepared very well, and it brings very important issue. I haven't noticed any problems that should be addressed in revision. Only I can recommend to add reference in line 99 (few studies have been conducted on the ability of different types of UBGLs) and to elaborate landscape pattern indicators which were selected (line 221). Although they are mentioned in table 4, I suggest to put it in 2.3.4. Ch. Also, title of the MS is not the same in the system and in the PDF.

Author Response

Point 1: Elaborate landscape pattern indicators which were selected (line 221). Although they are mentioned in table 4, I suggest to put it in 2.3.4. Ch

Response 1: I created Table 2 to illustrate the landscape pattern indicators. Table 2 is reflected in line 339.

Reviewer 2 Report

Dear authors, please find my review comments below. I have no general comments, but a few minor that are mostly related to clarity.

Comment #1: line 18

Please add full reference for LST (land surface temperature). This becomes clear only latter in the paper.

Comment #2: lines 135-136

"... will have a permanent population ... by 2021". Since this has passed rephrase the sentence so that reflects past information.

Comment #3: line 161

It is not clear to me what you mean by LST/UHII spatial distribution. Do you mean the land surface temperature of individual heat islands, or something else. If so, please rewrite it in a more simple way.

Comment #4: line 203

What are non-blue-green space (NBGS)? Something else than what is already in other categories? Add explanation in parenthesis.

Comment #5: line 208

Can you add once a more general definition of a "mesh size". It is not there.

Comment #6: line 215

What do you mean by "coordination between the CESL and CEDL"? This was not clear to me.

Comment #7: lines 221-222

Please add explanation of abbreviations PD, LSI, AREA_MN, etc. Some are given much latter in Table 4, which is impractical for the reader to find.

Comment #8: lines 229-230

What are signs in parenthesis for? (+, -) I cannot find the reason for that.

Comment #9: line 234

Replace "means" with "indicates" as it is in other tables.

Comment #10: line 237

Which are socioeconomic levels? What exactly do they refer to?

Comment #11: Why is regional financial strength important for predicting the demand for cooling? Please add an argument.

Comment #11: lines 249-250

You are referring to three main and eight secondary indicators, but then only 8 are listed. This was a bit unclear. I know they are in Table 4, but it would be better to have enough information in the first mentioning.

Comment #12: equation 5

Instead of N+ there should be N-.

Comment #13: equation 7

I cannot find an explanation of "p", or did I miss it.

Comment #14: lines 294-297

The reason for square parenthesis? I was not sure.

Comment #15: 307

I would change "worst" into "most close to zero". Is that what you meant?

Comment #16: line 317

Pearson is misspelled.

Comment #17: line 318

Could you replace "landscape component proportion" with "percentage of land-use"? If it makes sense.

Comment #18: Table 6

Why do you use "hm2". I would just use km2 or hectares. And, please add some explanation on the "Ratio (%)". Do you mean % of entire area?

Comment #19: Table 7

What do you mean by "Hysteretic"? Like lag of response?

Reviewer 3 Report

Overall, I like the article, but I have a few comments.
1. It is very difficult to read and understand the use of a large number of abbreviations, it is quite a common practice, but I would advise you to limit it. Abbreviations should absolutely not appear in the abstract - it is illegible.
2. Due to the length of the article, please consider to describe Fig. 2 in more detail in the methodology chapter, add full names in addition to abbreviations, and briefly describe "arrows" - in my opinion, this drawing should allow you to understand the entire topic of the work. The text is detailed explanations.
3 Conclusion 2 is not very understandable, too many mental abbreviations were used.

Reviewer 4 Report

The reviewed article touches on a very important problem, which is the need to improve the living conditions of city dwellers by improving microclimatic conditions. The article was written in understandable language. The methodology was presented in a very precise but clear way for the reader. The graphics also deserve praise. All drawings are legible and complement the text very well. Despite the generally positive reception, I have a few remarks which, if included in the final version of the article, may further improve its quality.

1. It seems to me that the introduction is too long, while the goal is not entirely clear.

2. Please leave a space between the subsection titles and the text. At the moment, it is merging into one in several places.

3. The methodological part lacks information on statistical analyses. Please explain what software was used to calculate the Pearson correlation coefficient values.

4. Lines 132-133 - information on climate requires an indication of source materials.

5. Lines 135-136 - information on the size of the population requires the indication of source materials.

6. Tables 2,3,4 - it seems to me that it is necessary to provide under the tables what the individual symbols that appear in them mean.

7. Table 6 - please change the area units to more commonly used “ha”.

8. Conclusions are too elaborate. Please answer the questions posed in lines 113-116 more clearly.
